# Control of Laves Precipitation in a FeCrAl-based Alloy Through Severe Thermomechanical Processing

**DOI:** 10.3390/ma12182939

**Published:** 2019-09-11

**Authors:** Jiyun Zheng, Yuzhen Jia, Peinan Du, Hui Wang, Qianfu Pan, Yiyong Zhang, Chaohong Liu, Ruiqian Zhang, Shaoyu Qiu

**Affiliations:** Science and Technology on Reactor Fuel and Materials Laboratory, Nuclear Power Institute of China, Chengdu 610213, China; zhengjiyun@aliyun.com (J.Z.); jaja880816@aliyun.com (Y.J.); dupeinan@126.com (P.D.); panqianfu1234@163.com (Q.P.); ustbzhangyiyong@163.com (Y.Z.); lch85904239@163.com (C.L.); fmkl@npic.ac.cn (S.Q.)

**Keywords:** FeCrAl alloy, Laves phase, precipitation particle, strain-induced precipitation

## Abstract

In recent years, the development of nuclear grade FeCrAl-based alloys with enhanced accident tolerance has been carried out for light water reactor (LWR) fuel cladding to serve as a substitute for zirconium-based alloys. To achieve excellent microstructure stability and mechanical properties, the control of precipitation particles is critical for application of FeCrAl-based alloys. In this paper, the effect of thermomechanical processing on the microstructure and precipitation behavior of hot-rolled FeCrAl alloy plates was examined. After hot rolling, the FeCrAl alloy plates had typical deformation textures. The rolling direction (RD) orientation gradually rotated from <100> to <110> along with increasing reduction. Shear bands and cell structures were formed in the matrix, and the former acted as preferable nucleation sites for crystallization. Improved deformation helped to produce strain-induced precipitation. The plate with 83% reduction had the most homogeneous and finest precipitation particles. Identification results by TEM indicated that the Laves precipitation was of the Fe_2_Nb-type crystal structure type, with impurities including Mo, Cr, and Si. The plate with uniform Laves particles displayed favorable heat stability after a long period of aging at 800 °C. The microstructure evolution of the aged sample was also observed. The deformation microstructure and the strain-induced precipitation mechanism of FeCrAl alloys are discussed.

## 1. Introduction

Zirconium alloys are a practical cladding material and are often used in the cores of light water nuclear reactors. However, under a loss of coolant accident (LOCA), a rapid production of heat and water steam with increasing temperature will occur in the core of nuclear reactor [1,2,3,4]. It has been demonstrated that zirconium alloy cladding severely reacts with steam at temperatures over 1200 °C, generating a mass of heat and hydrogen gas. The rapid generation of heat and hydrogen greatly exacerbates the degradation phenomena, resulting in the melting of the core and even a heavy explosion accident [5]. To enhance safety margins and reduce the critical heat removal limit from the reactor during the accident scenario, accident tolerant fuel (ATF) materials that exhibit significantly slower oxidation kinetics in high temperature steam environments have been proposed for the substitution of zirconium alloys [6,7].

FeCrAl-based ferritic alloys that contain significant quantities of Cr (~10–13 wt.%) and Al (~4–6 wt.%) are one of the most prospective cladding materials for ATF, due to their enhanced oxidation and corrosion resistance in water steam environments at temperatures over 1000 °C [3,8]. The excellent oxidation resistance of FeCrAl alloys originates from the abundant Cr and Al elements, which help to generate protective oxidation films (Cr_2_O_3_ and Al_2_O_3_) on the surface of the FeCrAl alloys, hindering further reactions between the Fe matrix and steam. It has been demonstrated that the Al_2_O_3_ film can be stable up to 1400 °C, which will extremely improve the oxidation resistance of cladding under the normal operation conditions or under a LOCA [9]. Besides, microalloy elements such as Mo and Nb can be added to improve the mechanical properties and microstructure stability of FeCrAl alloy cladding when used in a high temperature environment [10,11]. Currently, one of the main works in this field is focused on composition optimization, in order to maximize the aqueous corrosion resistance, high-temperature oxidation resistance, and mechanical properties of FeCrAl-based alloys, while FeCrAl alloys still maintain reasonable performance in terms of their processability, neutronics economy, and radiation tolerance.

To obtain excellent deformability (final size of FeCrAl cladding: 9.5 mm in outer diameter and ~0.35–0.37 mm in thickness) and various mechanical properties (such as yield strength, fatigue, and creep deformation resistance), microstructure control is key for FeCrAl-based alloys, and the control of Laves phase particles is the main focus here [12,13]. In FeCrAl alloys, fine and uniformly distributed Laves particles can result in precipitation strengthening and enhance the microstructure stability and mechanical properties [10]. However, due to the high Al and Cr content, FeCrAl alloys have a body-centered cubic (BCC) structure up to their melting temperature, eliminating the chance of grain refinement and solution by the phase transformation that occurs in Zr alloys and carbon steels. A similar feature is observed in other ferrous alloys, including high-Cr ferritic stainless steels and low-density FeAl alloys [14,15]. Besides, it has been reported that the FeNb Laves phase is the main precipitate in FeCrAl alloys with a solute Nb element [3]. According to the FeNb binary phase diagram, the Laves phase has a high heat stability that can hold up to 1200 °C, which means that the coarse Laves phase formed during solidification is hard to eliminate [16]. These factors lead to difficulties in the control and regulation of precipitation particles in FeCrAl alloys.

This paper is focused on the microstructure control and mechanical properties of FeCrAl alloys with the element Nb, especially in terms of the control of the size and distribution of Laves particles. The study helps us to understand the formation of the fine and uniformly distributed Laves precipitate and its effect on the mechanical properties of advanced FeCrAl alloys. Based on the results, a detailed discussion is carried out to produce further insight in the preparation and development of FeCrAl alloys to be used as ATF cladding material.

## 2. Experiment

The FeCrAl-based alloys prepared by vacuum induction melting and hot forging were processed by the wire-electrode method into sizes of 80 × 30 × 5 mm^3^. The nominal composition of the FeCrAl-based alloy is listed in Table 1. Then, the FeCrAl alloy plates were treated at 1300 °C for 2 h to realize a solution of alloy elements for the subsequent hot rolling. The FeCrAl alloy plates were pre-heated to 780 °C for 2 h and deformed to three nominal reductions of thickness: 50%, 70%, and 90%. After hot-rolling, the FeCrAl alloy plates were continuously annealed at 780 °C for 1 h. To study the thermal stability, the hot-rolled plates were annealed at 800 °C for up to 125 h.

The microstructures of the manufactured or annealed samples were characterized using an optical microscope (OLYMPUS OLS4000, Olympus Corporation, Tokyo, Japan), a scanning electron microscope (FEI Nova Nano SEM-400, FEI Company, Hillsboro, OR, USA), electron backscattered diffraction (EBSD, NordlysMax2, Oxford Instruments, Oxford, UK), and a transmission electron microscope (TEM, Tecnai G2 F20, FEI Company, Hillsboro, OR, USA). Metallographic samples were mechanically polished and then etched in a solution of FeCl3, HCl, and deionized water. The SEM and EBSD samples were electropolished in an 8 vol.% perchloric acid and alcohol mixed liquid. For the deformed samples, the scanning location of all specimens was near the surface area of the transverse direction (TD) surface, containing the rolling direction (RD) and the normal direction (ND). The EBSD data, with proper noise reduction, were analyzed by a channel 5 software. To exhibit the orientation relationship between the microstructure and macroscope direction, an inverse pole figure (IPF) map was represented. The colors in IPF maps represent corresponding crystallographic orientations relative to the rolling direction of the hot-rolled FeCrAl alloy plates based on the orientation legend. Boundaries with a misorientation larger than 10° (high-angle) were superimposed on the IPF maps as black lines. The morphology and distribution of precipitation particles were observed by the SEM pictures captured on the ND surface. The TEM samples were prepared by a twin-jet electropolisher (Struers Tenupol-5, Struers, Ballerup, Denmark) and observed to obtain the morphology, diffraction pattern, and chemical composition of the precipitation particles. Tensile tests were performed along the RD direction at room temperature using a universal tensile machine. The gauge size of sub-sized sheet tensile specimen (a dog-bone shape) was 1.5 mm in width and 5 mm in length. The strain speed was constant at 1 × 10−4 s−1 by cross head control. Three parallel tensile samples were prepared by 800 grit SiC grinding paper and then tested for each processing condition. The microhardness of the annealed specimen was examined with a HV microhardnesss tester (FM-700, Minsks Company, Xi’an, China). The dwell time and load used for microhardness testing was 10 s and 100 g, respectively. The annealed specimens were then observed by SEM to reveal the change of the microstructure.

## 3. Results

Figure 1 shows the optical microscope images of the FeCrAl alloy after solution treatment. It is clear that the FeCrAl alloy is free of precipitates, both at grain interface and inner grains after solution treatment. The actual thickness reduction of hot-rolled FeCrAl alloy plates was tested, as shown in Table 2. The actual deformation of FeCrAl alloy plates was 52%, 67% and 83%, respectively. In the following part, symbols S1, S2, and S3 are used to represent the three FeCrAl plates with different deformation.

Figure 2 displays the EBSD IPF maps of the FeCrAl alloy plates, with different hot-rolling reduction results. The orientation relative to the RD is based on the orientation legend inserted in Figure 2. Many distinct bands with widths of a few micrometers, as marked by the solid white line in Figure 2, were formed in matrix. These bands had a tilting angle range from 20° to 45° inclining to the rolling direction. It should be noted that the overall tilting angle in S3 was lower than that in S1 and S2. Besides, with the increase in reduction, the orientation of the deformed matrix rotated mainly from <100>∥RD to <110>∥RD. Due to the extreme deformation, the S3 matrix displayed an orientation with network construction (<110> or <111>∥RD). On the other hand, well-defined micro-grains, enclosed by the boundary, with a misorientation larger than 10° were formed in the shear band, while the inner misorientation was small, as shown in Figure 3a. The high angle boundary with a typical misorientation of the recrystallization boundary had a misorientation angle of 10°–60°, which divided the shear band into many “polygonised grains”. However, the misorientation of the interior grains was small with average misorientation of 0.54°. The size of the well-defined “polygonised grains” was ~2–5 μm, which is close to the threshold value of the recrystallized grains. Considering the micro size and distinct grains in the shear band, it is inferred that the shear band acts as the preferred site for recrystallization nucleation. Although recrystallization nuclei were found in the matrix, the overall recovery of the deformation microstructure did not occur. Other than shear bands, a deformation cell structure was also found in the matrix, as marked by the dashed white line in Figure 2. The deformation cell structure, with a size of 1–2 μm, intersected with the shear band and had an alternative orientation. Figure 3 presents the point-to-point misorientation along the line in the selected rectangle areas of Figure 2. The misorientation between the deformation cell band structure was small, approximately 1°–2°, as shown in Figure 3b. A similar microstructure of the deformation cell band was also observed in other materials [17]. However, a large misorientation boundary was not observed in the region of the deformation cell band, which implies more homogeneous deformation than that in the shear band.

Figure 4 presents the observation of precipitation particles in the matrix of the hot-rolled FeCrAl alloy plates. After hot rolling, a mass of precipitation particles was formed in the matrix of the FeCrAl alloy plates. Compared with the original solution state in Figure 1, the thermomechanical process by hot rolling accelerated the precipitation of the solution alloy elements. However, it is clear that the morphology of the precipitation particles was extremely modified with the increase of rolling reduction. Basically, there were two kinds of precipitation particles in the matrix of S1 and S2: Granular, fine nano-particles with a high density and coarse micro-particles with a low density, which partially aggregated in clusters, as marked by the arrows and rectangle regions in Figure 4. It is interesting that the area located by the coarse micro-particles had distinguished cellular grains with clear contrast, whereas that located by the fine nano-particles displayed an unclear sub-microstructure, as shown in Figure 4a. This difference indicates the effect of microstructure stability caused by fine precipitation particles with a nanoscale size. Sun pointed out that the microstructure stability of FeCrAl alloys was aroused by the fine nano-particles, which effectively pinned the movement of sub-boundaries at high temperature [10]. With further increasing deformation reduction, S3 shows a totally different microstructure, where only fine and uniformly distributed precipitation nano-particles were formed throughout the matrix, although a few particle clusters were found. The statistical size of the fine nano-particles in each specimen was 73.6 ± 42.0 nm, 86.0 ± 48.7 nm and 123.5 ± 65.5 nm, respectively. Although a smaller size of fine nano-particles was obtained in S1 and S2, the formation of coarse micro-particles and clusters indicates ineffective control of the size of precipitations. The result shown in Figure 4 demonstrates that the size and distribution control of precipitation particles can be realized by hot processing through increasing deformation strains of the FeCrAl alloy.

Figure 5 displays the representative engineering stress-strain curves of FeCrAl alloy plates with different amounts of hot-rolling reduction. Table 3 shows the summarized results of the mechanical properties. With the increase of deformation strain, the yield and ultimate tensile strength increased, while the elongation decreased. Although the FeCrAl alloy plates were held at 780 °C for 1 h to eliminate residual deformation stress after hot rolling, the effect of work hardening on the tensile property cannot be ignored. On the other hand, the fine nano-particles pinned the movement of dislocation and delayed the beginning of material yielding under an applied load. The uniform distribution of precipitation particles was beneficial for the improvement of the strengthening effect [18]. Considering the uniform distribution of precipitation particles and a larger rolling reduction, both work hardening and precipitation strengthening promoted the tensile strength of S3.

Figure 6 displays the TEM images and identification results of precipitation particles in S3. As shown in Figure 6a, sub-grains with a size of ~500 nm were formed. Although a few particles were located in the interior area of the sub-grains, most of the particles were mainly precipitated at the sub-grain boundary, which indicates that the nucleation of precipitation particles is closely related to the dislocation density. The sub-grain boundary, with a small misorientation angle, contains many arrangements of dislocations, which act as preferential locations for second precipitation particles [19]. Besides, deformation at high temperature further accelerated the diffusion of the solution alloy element. Therefore, the uniform distribution of fine nano-particles was generated throughout the matrix. Basically, three kinds of precipitation particles were found in the matrix: Spherical (the most common), clubbed, and cluster, shown in Figure 6b–d, respectively. The diffraction pattern images were also inserted in Figure 6. Although the morphology of the second particles was different, the identification results prove that the crystal structure of all precipitation particles was the Fe_2_Nb compound. Table 4 presents the composition of three types of Laves phase particles. The composition consisted of the elements Fe, Nb, Mo, Si, Cr, and Al. Besides, the atom ratios of Fe/(Nb + Mo) were close in each particle, namely 1.68, 1.88, and 1.76, respectively. The composition results confirmed that the precipitation particle was a Fe_2_Nb-type Laves phase, with impurities including Mo, Cr, and Si atoms. This result is in accordance with the observation of Sun [10], who indicated that the precipitation particles contained many additional elements, like Mo/Cr/Al/Si, in thermomechanical-processed FeCrAl alloy plates. The results in Figure 6 prove that a complicated Laves Fe_2_Nb-type compound with other alloy elements preferentially formed at the boundary. The pinning effect of stable Laves particles helps to improve the microstructure stability of FeCrAl alloys at high temperature.

To evaluate the thermal stability of hot rolled FeCrAl alloy plates, samples S1 and S3 were annealed at 800 °C for different periods of time to reveal the change in hardness and microstructure. Figure 7 shows the Vickers hardness of the hot-rolled FeCrAl alloy plates, plotted as a function of their aging time. The hardness of S1 and S3 gradually decreased with holding time at 800 °C. Sample S1 presented a significant drop in hardness at the early stage of annealing time (below 5 h), while the hardness of S3 inversely increased during this period. Beyond 5 h, hardness basically decreased with annealing time. Once the time was over 40 h, the hardness change during further annealing was not obvious. It is clear that the hardness of S3 was larger than that of S1, indicating the higher microstructure stability of S3. Figure 8 shows the Laves particle morphology of the hot-rolled FeCrAl alloy plate S3 with different annealing times. The original Laves particles displayed a close to spherical shape. When the annealing time was 5 h, the precipitation particles became totally round in their morphology, with a uniform size. The matrix still presented a deformation state with indistinct contrast, as shown in Figure 8b. However, long time holding up to 125 h led to the severe solution of fine Laves particles and the remaining particles were coarsened to a large size. Besides, the solution and coarsening of Laves particles deteriorated the microstructure stability of the FeCrAl alloy. The matrix presented an obvious contrast between sub-grains, which indicates sufficient recovery, as shown in Figure 8c. The result shown in Figure 8 emphasizes the importance of fine and uniformly distributed Laves particles for ensuring the microstructure stability of the FeCrAl alloy.

## 4. Discussion

### 4.1. Deformation Microstructure and Preferential Recrystallization

As demonstrated in Figure 2, the hot rolled FeCrAl alloy plates show two different kinds of deformation microstructure: Cell band and shear band, which have been widely reported in face cubic crystal (FCC) materials such as aluminum, copper, and nickel [20]. The cell band presents a slightly elongated sub-microstructure or sometimes equiaxed cells, with quite well-defined boundaries, mainly lying with an angle referring to the rolling direction. Hansen [21] describes the features of the cell band as lamellar boundaries, which are also referred to as geometry necessary boundaries (GNBs), consisting of orderly arranged dislocation walls in the interior of each band. The misorientation between cell boundaries is about 2°–5°, while that between the cell walls is usually less than 1°, as shown in Figure 3. However, the other kind of band is frequently intersected by elongated sub-microstructures at an angle of about 30° to the main family of the lamellar. Such an intersecting shear band is also called an S-band, in which the misorientation is significantly larger than that in the lamellar substructures, probably reaching 10° or more. Hurley [17] indicates that the cell band structure is a result of the dynamic recovery of the dislocation debris produced during slip, in which the dislocation walls are continually dissolved, reformed, and realigned. On the other hand, however, the shear band may derive from a geometric instability resulting from the inability of the material to sustain further work hardening [22].

It is worth noting that some recrystallized micro-grains in the shear band showed an orientation inclining to the <111> direction, as shown in Figure 2. In fact, the grain shear band plays an important role in the nucleation of <111>∥ND in similarly warm-rolled Interstitial Free (IF) steel. The deformed grains with a <111>-γ fiber orientation are identified as preferential nucleation sites due to their higher stored energy [23,24]. It is reported that grains with large Taylor factors suffer high levels of slip activity during deformation and accumulate the largest dislocation densities and stored energy, although this is not homogeneous [25]. Due to higher Taylor factor, the stored energy in <111>-γ fiber is higher than that in other typical fibers [26]. The well-defined micro-grains with γ fiber orientation shown in Figure 2 indicate nuclei at the first period of recrystallization and inhomogeneous nucleation process.

### 4.2. Strain-Induced Precipitation

The strain-induced precipitation has been widely researched and also applied for the microstructure control for many microalloyed steels or aluminum alloys [25,27]. Dutta et al. describe the activation energy for heterogeneous nucleation as follows [28]:(1)$$\Delta G\;=\;\Delta G_{chem}\;+\;\Delta G_{int}\;+\;\Delta G_{disl}\;=\;-\frac{4}{3}\pi R^{3}\Delta G_{v}\;+\;4\pi R^{2}\gamma \;-\;0.4\mu b^{2}R$$
where Δ*G* is the total free energy, Δ*G_chem_* is the chemical free energy, Δ*G_int_* is the interfacial energy, *R* is the particle radius, Δ*G_disl_* is the dislocation core energy over the precipitate radius, Δ*G_v_* is the change of volume free energy, *γ* is the surface energy of the precipitate, and *μ* and *b* are the shear modulus and Burgers vector, respectively. In addition, the critical radius for nucleation is given as [28],
(2)$$R_{c}\;=\;-\frac{2\gamma }{\Delta G_{v}}$$

Replace the radius *R* by the critical radius *R_c_*, and activation energy for nucleation can then be obtained as [28],
(3)$$\Delta G^{*}\;=\;\frac{16}{3}\pi \frac{\gamma^{3}}{\Delta G_{v}^{2}}\;-\;0.8\mu b^{2}\frac{\gamma }{\Delta G_{v}}$$

This equation indicates that the activation energy decreases as a particle is nucleated on dislocation because of the negative item, which results in the heterogeneous nucleation of precipitation on dislocations.

On the other hand, the nucleation rate (number of precipitates per unit volume and time) is expressed as [28],
(4)$$J\;=\;N_{0}Z\beta^{\prime }exp\left(-\frac{\Delta G}{kT}\right)exp(-\frac{\tau }{t})$$(5)$$\beta^{\prime }\;=\;\frac{4\pi R_{c}^{2}DC_{M}}{a_{0}^{4}}$$(6)$$N_{0}\;=\;0.5\rho^{2/3}$$
where *J* is the nucleation rate, *Z* is the Zeldovich factor (~1/20), *β’* is the atomic impingement rate, *τ* is the incubation time, *k* is Boltzmann constant, *N*_0_ is the number of heterogeneous nucleation sites, *D* is the bulk diffusivity, *a*_0_ is the lattice parameter, *C_M_* is the concentration of alloy element, and *ρ* is the dislocation density. The above equations reveal that the increase of dislocation density effectively improves the nucleation rate and leads to faster nucleation. It is usually assumed that the incubation time *τ* of precipitation is close to zero when there exists a high density of dislocation. In addition, the diffusion of solute atoms in matrix can be enhanced by increasing the dislocation density, due to the lower activation energy of pipe diffusion than that of bulk diffusion. Besides, the dislocation density of metal materials could be dramatically increased through large deformation, resulting in increasing the amount of nucleation sites for precipitation [29]. Therefore, sample S3 obtained a fine and uniform distribution of Laves precipitate when compared with that in relative low strain samples.

However, when the total deformation was lower than 70%, some coarse precipitation particles were formed in the matrix. It is interesting that the number of coarse precipitation particles gradually decreased with the increase of deformation strain. Based on deformation theory, the overall deformation throughout the matrix is not uniform due to different crystal orientations. According to the Schmid law, the grains with a large Schmid factor easily deform under the same amount of loading [30]. Therefore, the inhomogeneity of deformation at the micro-crystal level causes the dislocation sites for precipitation to not be uniform. It was reported that the shear band with higher dislocation density in low carbon steels preferentially occurred in grains with larger *M* values [23]. The heterogeneous deformation at the micro-level responds to the inhomogeneous precipitation of Laves phase at low strain.

## 5. Conclusions

The effect of thermomechanical processing on the precipitation behavior of FeCrAl-based alloy during hot rolling was discussed in this paper. The conclusions are listed as follows:The FeCrAl alloy plate had typical deformation microstructure, with a mix of shear band and cell band structure after hot rolling. The orientation of the matrix gradually rotated from <100>∥RD to <110>∥RD with increasing reduction. The shear band was oriented at ~20°–45°, referring to the processing direction, and acted as preferable nucleation sites for precipitation.Improved deformation contributed to causing strain-induced precipitation. The FeCrAl alloy plate with 83% reduction had the most homogeneous precipitation particles. The identification results by TEM indicated that the Laves phase precipitate was Fe_2_Nb-type crystal structure with impurities including Mo, Cr, and Si atoms.The FeCrAl alloy plate with uniform Laves particles displayed favorable heat stability after a long period of aging at 800 °C. However, the aging effect of Laves particles was obvious up to 125 h of holding. The microstructure stability of FeCrAl alloy was closely related to the distribution of fine Laves precipitates. The strain-induced precipitation mechanism could be taken use of for the preparation of FeCrAl alloy cladding with fine and uniform precipitation.

## Figures and Tables

**Figure 1 materials-12-02939-f001:**
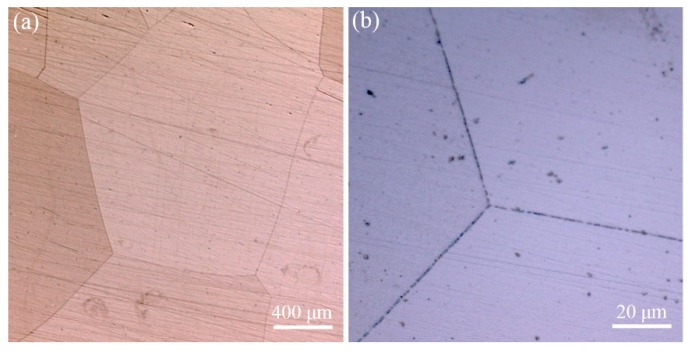
Optical microscope images of FeCrAl alloy after solution treatment at 1300 °C: (**a**) a low magnification image and (**b**) a higher magnification image.

**Figure 2 materials-12-02939-f002:**
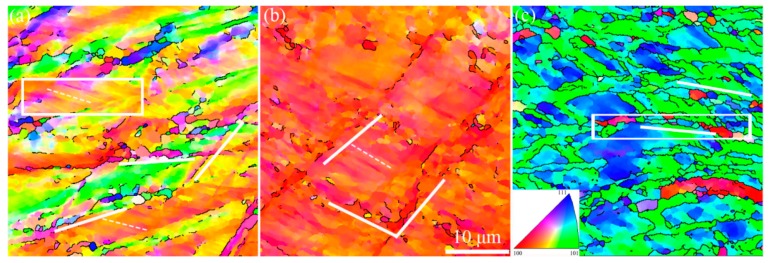
Electron backscattered diffraction (EBSD) inverse pole figure (IPF) maps (relative to the rolling direction, RD) of FeCrAl alloy plates with different hot-rolling deformations: (**a**) S1, (**b**) S2, and (**c**) S3. The inserted picture is the orientation triangle map. Black lines represent boundaries with a misorientation larger than 10°. Dashed and solid white lines represent the deformation cell band and shear band, respectively.

**Figure 3 materials-12-02939-f003:**
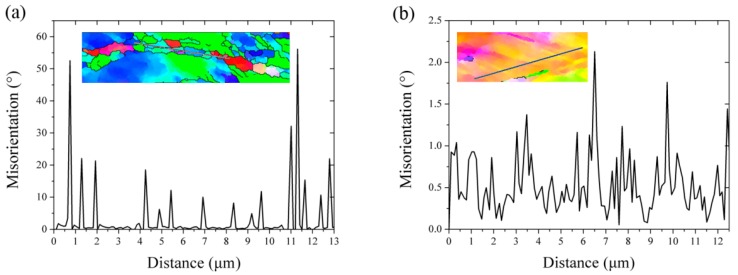
The point-to-point misorientation along the line in selected rectangle areas of Figure 2: (**a**) S3 and (**b**) S1.

**Figure 4 materials-12-02939-f004:**
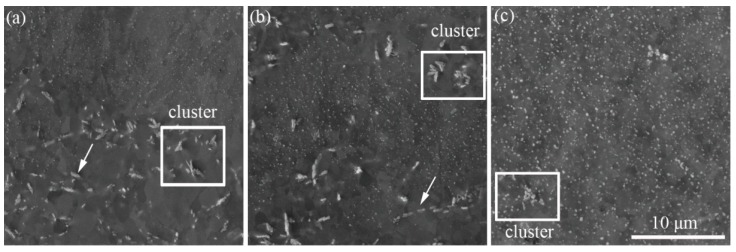
SEM pictures of FeCrAl alloy plates with different hot rolling reduction: (**a**) S1, (**b**) S2 and (**c**) S3. The arrow marks coarse particle and rectangle emphasizes the particle cluster.

**Figure 5 materials-12-02939-f005:**
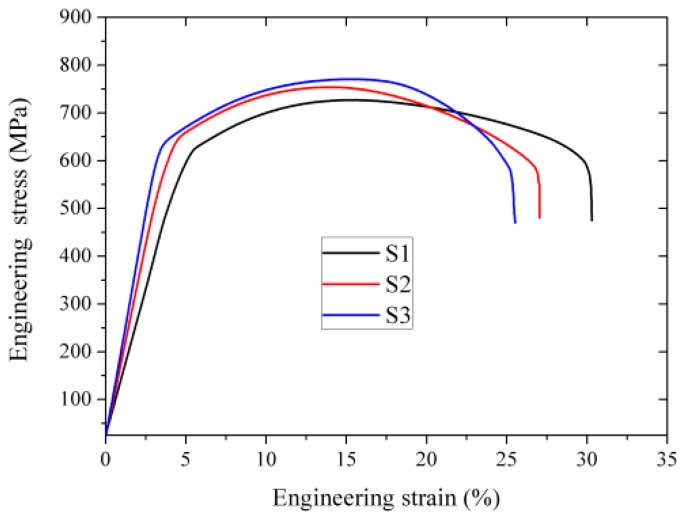
Representative engineering stress-strain curves of FeCrAl alloy plates with different hot rolling reduction.

**Figure 6 materials-12-02939-f006:**
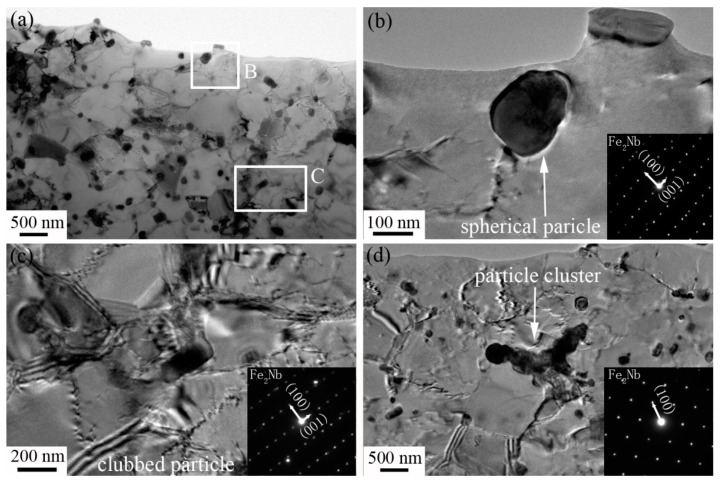
TEM observation images and identification results of precipitation particles in S3 for (**a**) overall image, (**b**) region B in (**a**), (**c**) region C in (**a**) and (**d**) precipitation cluster. The inserted pictures are corresponding diffraction patterns.

**Figure 7 materials-12-02939-f007:**
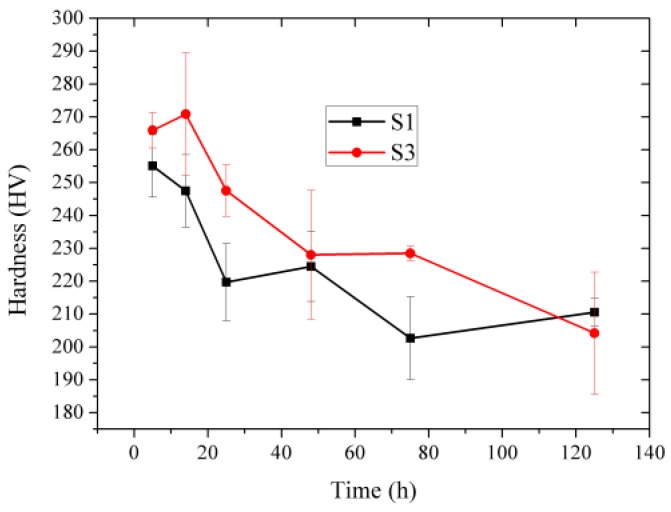
Vickers hardness of hot-rolled FeCrAl alloy plates plotted as a function of aging time at 800 °C.

**Figure 8 materials-12-02939-f008:**
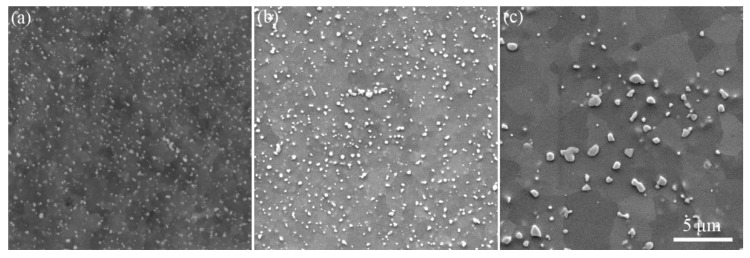
SEM images of S3 with different annealing time: (**a**) Original state, (**b**) 5 h, and (**c**) 125 h.

**Table 1 materials-12-02939-t001:** Nominal composition of the FeCrAl-based alloy (wt.%).

Fe	Cr	Al	Mo	Nb	Ti	V	Y
bal.	13	4.5	2	3	0.1	0.1	0.05

**Table 2 materials-12-02939-t002:** The actual thickness reduction and deformation for hot-rolled FeCrAl alloy plates.

Nominal Deformation	Thickness after Hot Rolling	Actual Deformation	Symbol
50%	2.4	52%	S1
70%	1.65	67%	S2
90%	0.85	83%	S3

**Table 3 materials-12-02939-t003:** Mechanical property results of FeCrAl alloy plates (at least 3 parallel specimens).

Samples	YTS ^a^/MPa	UTS ^b^/MPa	Elongation/%
S1	560.8 ± 10	733.4 ± 18.2	34.2 ± 5
S2	589.9 ± 11.8	738 ± 22.9	26.9 ± 0.8
S3	593.9 ± 18.9	747.3 ± 20.9	20.6 ± 3.6

^a^ Yield tensile strength; ^b^ ultimate tensile strength.

**Table 4 materials-12-02939-t004:** Composition of three type precipitation particles identified by TEM (atom fraction, %).

Particle Type	Fe	Nb	Mo	Si	Cr	Al	Atom Ratio (Fe: Nb + Mo)
Spherical	48.78	13.19	15.06	9.63	7.95	2.16	1.68
Clubbed	51.5	14.2	13.12	8.04	8.30	2.37	1.88
Cluster	49.73	14.4	13.82	11.44	8.07	2.52	1.76

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
