# Peer review of "Control of Laves Precipitation in a FeCrAl-based Alloy Through Severe Thermomechanical Processing"

_materials, 2019, doi:10.3390/ma12182939_

Round 1

Reviewer 1 Report

The paper covers an interesting and important study. The data are well presented and the discussion is mostly based on the results. There are a few issues (Elastic modulus, hardness over time and thermal stability) to improve, but most of the comments are minor facts.

please check the text for blank spaces and English language...line 31 …core of the reactor… in 2. Experiment: Table 1 and Figure 1 are Results and should be described in 3. Results line 86: the contrast in Figure 1 needs improving line 87: Figure caption Figure 1: in both images (a) and (b) grains with grain boundaries are seen, it would be wise to say, that there is a higher magnification in (b) line 177: Elasticity modulus: I guess you mean the “Elastic Modulus”, “Modulus of Elasticity” or better “Young’s Modulus” You provide 12.8 to 19.3 GPa for the Elastic Modulus, however – this data should be much larger (around 200 GPa) Line 172: please rather say “cannot” instead of can`t Line 184: Table 3: please call the parameter rather TYS (tensile yield strength), UTS (ultimate tensile strength) and Elongation Line 126: just say Vickers hardness (in experimental you give the load, so it is known that it is Micro) Line 218 and 220: not sure, if “sudden drop” and “minor” are the correct words to say, the sudden drop is about 25HV and minor is around 15HV. I would instead of sudden rather say significant. Line 232: Figure 7 – there is a shift of the data to the right, by 5h – please line up the data to the correct times Line 311: still wondering about the large Elastic Modulus, values seem too small Line 313: can you really summarize to have a heat stability after long time aging, there are no data shown after 120h, rather say: up to 120 h. And a change from 260HV to 200HV is also not really stable – please modify that statement

Reviewer 2 Report

Row 143: Add a description for Figure 3 (a) and 3 (b)

Move rows 210 - 213 to row 188

Move rows 232 - 234 to row 214

Row 321: J. Nucl. Mater. 2013,440, 420-427. => J. Nucl. Mater. 2013, 440, 420-427.
